# Evolution of Neo-RAS-WT in Circulating Tumor DNA from First-Line to Subsequent Therapies in Metastatic Colorectal Cancer

**DOI:** 10.3390/cancers17071070

**Published:** 2025-03-22

**Authors:** Marco Siringo, Michela De Meo, Irene Bottillo, Paola Grammatico, Enrico Cortesi, Chiara Nicolazzo, Paola Gazzaniga

**Affiliations:** 1Department of Molecular Medicine, Sapienza University of Rome, 00185 Rome, Italy; marco.siringo@uniroma1.it (M.S.); michela.demeo@uniroma1.it (M.D.M.); 2Division of Medical Genetics, Department of Experimental Medicine, San Camillo-Forlanini Hospital, Sapienza University, 00185 Rome, Italy; irene.bottillo@uniroma1.it (I.B.); paola.grammatico@uniroma1.it (P.G.); 3Fondazione per la Ricerca Oncologica (FORO), 00196 Rome, Italy; 4Department of Life Science, Health and Health Professions, Link Campus University, Via del Casale di San Pio V, 00165 Rome, Italy; c.nicolazzo@unilink.it; 5Department of Experimental Medicine, Sapienza University, 00161 Rome, Italy

**Keywords:** liquid biopsy, colorectal cancer, RAS

## Abstract

In RAS mutant colorectal cancer, liquid biopsy has enabled the identification of “neo-RAS-Wild-Type”, a transient phase characterized by the disappearance of RAS mutations, with significant clinical implications for re-sensitization to EGFR blockade. A prospective study was designed to track neo-RAS-WT in plasma samples from patients with RAS-mutant colorectal cancer across three different lines of treatment. We demonstrated that most neo-RAS-WT patients had sustained neo-RAS-WT status that could be confirmed at the time of disease progression, independent of treatment line. This emphasizes the importance of correctly differentiating neo-RAS-WT from non-shedding cases, offering new perspectives for EGFR blockade therapy.

## 1. Introduction

In metastatic colorectal cancer (mCRC), RAS testing is a critical part of treatment planning and is required before starting anti-EGFR therapies [1].

Due to the high concordance rate between the primary tumor and metastatic sites, RAS retesting is typically not performed in routine clinical practice. Nevertheless, the molecular diversity of mCRC indicates that cancer cells often undergo temporal heterogeneity under the selective pressures exerted by therapies [2,3].

Several studies have consistently shown the value of liquid biopsy analysis in tracking tumor evolution in patients with mCRC, offering prognostic insights and helping guide treatment decisions. Specifically, repeated analysis of circulating tumor DNA (ctDNA) in RAS wild-type tumors has enabled the identification of emerging drug resistance biomarkers and novel potential drug targets [4].

The term “neo-RAS-wild-type (WT)” describes a temporary phase in the clonal evolution of RAS-mutant mCRC, characterized by the disappearance of RAS mutation in ctDNA over time [5,6]. This temporary RAS-WT window, which relies primarily on liquid biopsy assessment, has garnered increasing interest in the clinic, suggesting that repeated RAS testing through liquid biopsy might change the selection of patients who are candidates for EGFR inhibitors (EGFRi).

Consistently, several phase II clinical trials are currently investigating the effectiveness of EGFRi in patients with neo-RAS-WT disease [7,8,9].

Despite growing clinical interest, the neo-RAS-WT concept remains a topic of debate. RAS mutations may become undetectable due to shifts in ctDNA release caused by effective treatment or, alternatively, tumor evolution. This phenomenon has been reported to occur in a highly variable proportion of cases (6.6–67%) across studies. Such inconsistency may arise from the lack of standardized methods to reliably distinguish ’neo-RAS-WT’ cases from tumors that do not shed ctDNA. In fact, a low burden of disease and encephalic or pleural progression generally characterize patients whose tumors release low quantities of ctDNA (“non shedding”), with an increased risk of possible false-negative results [5,10].

Thus, an erroneous interpretation of “neo-RAS-WT” could be caused by the lack of a “normalizer” of the quantity of ctDNA available in plasma. Furthermore, the inconsistency among different studies could be attributed to the different timings of ctDNA analysis for neo-RAS-WT detection. While some studies have reported a high rate of neo-at failure of first-line treatment, others have described a lower prevalence in subsequent treatment lines [11,12,13,14]. To date, the incidence of “neo-RAS-WT” over different treatment lines has not been prospectively investigated. This is the first prospective study aimed at assessing “neo-Ras-WT” in ctDNA across RAS-mutant mCRC patients receiving first-line and post-first-line systemic therapies.

## 2. Materials and Methods

Thirty-five (35) patients with RAS-mutated (MT) mCRC who were candidates for first-line treatment were prospectively enrolled. The inclusion criteria were as follows: histologically proven diagnosis of colorectal adenocarcinoma; RAS mutation detected in the primary tumor and/or related metastasis at the time of the initial diagnosis; RAS mutation detected in ctDNA at the time of initial diagnosis; age > 18 years; ECOG performance status ≤ 2; no previous treatment for metastatic disease (previous adjuvant treatment was allowed); and signed informed consent. The patients were treated with standard first-line chemotherapy with or without bevacizumab according to national guidelines. Following disease progression, the patients received second- and third-line treatments according to the investigator’s choice. Treatment continued until progression, death, withdrawal of consent, or unacceptable toxicity. Clinical response was evaluated according to the Response Evaluation Criteria in Solid Tumors, version 1.1 (RECIST 1.1). We collected longitudinal blood samples at baseline and every 3 months during each treatment line until disease progression.

The authorization used to perform liquid biopsies was released by the Regional Ethical Committee (No. 179/16), and this study was conducted in accordance with the Declaration of Helsinki. Plasma samples were obtained by centrifuging blood at 2000× *g* for 10 min, followed by removal of the plasma, which was then centrifuged again at 2000× *g* for 10 min. The plasma samples were stored at −80 °C until further use. ctDNA was extracted from up to 8 mL of plasma per patient via the Maxwell^®^ RSC 96 Instrument (Promega Corporation, Madison, WI, USA) with the Maxwell^®^ RSC ccfDNA LV Plasma Kit (Promega; cat. No. AS1840) according to the manufacturer’s instructions. Briefly, the plasma was transferred to a 50 mL tube, to which an equal volume of binding buffer and 100 µL of magnetic resin were added. The sample was incubated with shaking for 45 min and then centrifuged at 1000× *g* for 2 min to pellet the resin. The tube was placed in a magnetic rack for 1–2 min, and the supernatant was discarded. The resin was resuspended in the solution from well #1 of the Maxwell^®^ RSC cartridge, and the resuspended resin was transferred back to well #1 to initiate the extraction. The extracted DNA was quantified, and the index fragmentation was assessed via the EasyPGX qPCR instrument and EasyPGX analysis software (Diatech Pharmacogenetics, Jesi, AN, Italy https://www.diatechpharmacogenetics.com/linea-easy-pgx/, (accessed on 18 March 2025)), following the manufacturer’s protocol. A 25 µL aliquot of ctDNA was used to generate libraries with the Myriapod NGS Cancer Panel DNA Kit (Diatech Pharmacogenetics), which analyzes single-nucleotide polymorphisms and indels in 17 clinically relevant genes (ALK, BRAF, EGFR, ERBB2, FGFR3, HRAS, IDH1, IDH2, KIT, KRAS, MET, NRAS, PDGFRA, PIK3CA, POLE, RET, and ROS1). The samples were amplified via multiplex PCR via two primer mixtures to obtain 101 fragments (103–171 bases) covering the hotspot regions of interest. After purification with magnetic beads to remove residual primers, an indexing reaction was performed to add unique barcodes and sequencing adapters. Libraries were normalized by quantity using magnetic beads for consistent coverage. The normalized libraries were pooled and sequenced on the iSeq platform (Illumina Inc., San Diego, CA, USA). Data analysis was performed via Myriapod NGS Data Analysis Software (Diatech Pharmacogenetics https://www.diatechpharmacogenetics.com/linea-myriapod-ngs-dry/, (accessed on 18 March 2025)), which filtered out germline and low-quality variants.

Neo-RAS-WT was defined as the absence of RAS mutations in ctDNA after treatment initiation in patients who were diagnosed with RAS-MT (KRAS exons 2, 3, 4 or NRAS) upon pre-treatment tissue examination.

### Statistical Analysis

Continuous variables are reported as medians and ranges, whereas categorical variables are reported as counts and percentages. Progression-free survival (PFS) was defined as the time from treatment initiation until disease progression or last contact without disease progression, and overall survival (OS) was defined as the time between the start of first-line treatment and death from any cause or last contact. We estimated OS and PFS with Kaplan–Meier survival curves. The log-rank test was used to assess differences between groups. The correlations between categorical variables were analyzed via chi-square tests and binary logistic regression. Univariate and multivariate Cox proportional hazard models were employed to compare the WT window and survival outcomes. The median OS and PFS rates are reported with 95% confidence intervals (CIs). Statistical significance was defined at *p* ≤ 0.05. All the statistical analyses were performed via IBM SPSS Statistics version 23 (IBM, Armonk, NY, USA).

## 3. Results

### 3.1. Population Characteristics

A total of 35 patients with mCRC were enrolled, consisting of 20 men and 15 women. The median age at the time of inclusion was 61 years (range: 44–79 years). All patients met the inclusion criteria for RAS-mutant disease in both tumor tissues and ctDNA at baseline. Specifically, 32 patients (91%) had KRAS mutations (exon 2: 69%; exon 3: 6%; exon 4: 16%), while three patients (9%) harbored NRAS mutations. The tumor location was predominantly left-sided (74%), whereas 26% of the patients had right-sided tumors. The distribution of metastasis sites at diagnosis was as follows: 28 (80%) patients with liver metastases, 6 (17%) patients with peritoneal metastases, 12 (34%) patients with lung metastases, and 2 (5%) with bone metastases.

RAS status was prospectively monitored in longitudinal plasma samples across three consecutive treatment lines. A total of 380 serial blood samples were collected and analyzed, with each patient providing a median of 10 ctDNA samples (range: 8–14) at three-month intervals. A table presenting the type of RAS mutations detected on tissue and ctDNA is available in the Appendix A. For each treatment line, the patients were divided into the following three categories on the basis of whether RAS mutational status changed during the course of treatment: (1) neo-RAS-WT, defined as patients with no evidence of RAS mutation in plasma, but evidence of somatic mutations in genes other than RAS when reversions occurred; (2) non-shedding, defined as patients with loss of RAS mutation in plasma without concurrent somatic mutations in genes other than RAS; and (3) persistent mutant, defined as persistence of RAS mutation in plasma.

### 3.2. Longitudinal Monitoring of RAS Status in Plasma ctDNA During First-Line Treatment

Of the 35 patients enrolled, 24 (68%) transitioned to RAS-WT during first-line treatment. Among them, six patients were classified as neo-RAS-WT (group 1), confirmed by the detection of other somatic mutations in plasma samples, while 18 patients were classified as non-shedding (group 2). The remaining 11 patients were found to have persistent RAS mutations through serial ctDNA analysis (group 3). The rates of neo-RAS-WT, non-shedding and persistent mutant during first-line treatment are shown in Table 1.

In the six neo-RAS-WT patients, the initial RAS mutation became undetectable at 3 months in three patients (50%) and at 6 months in the remaining three patients (50%). All the new-RAS-WT patients had a sustained neo-RAS-WT status that could be confirmed at the time of disease progression. The median duration of the first-line neo-RAS-WT window was 5.5 months (range 3–9 months).

In the 18 non-shedding patients, the initial RAS mutation became undetectable at 3 months in four patients (22%), at 6 months in eleven patients (61%) and at 9 months in the remaining three patients (17%). The loss of RAS mutations in non-shedding patients was always transient since all patients reacquired RAS mutation status at the time of disease progression, with a return of the same variant in 13 patients (72%) and acquisition of a different mutation in 5 patients (28%). The median duration of the RAS-WT window in non-shedding was 4.7 months (range, 1–9 months).

At the time of progression from first-line therapy, 17% of patients were new-RAS-WT, and 83% were RAS-mutant (Table 1).

### 3.3. Longitudinal Monitoring of RAS Status in Plasma ctDNA During Second-Line Treatment

Before starting the second-line treatment, a further blood draw was performed in all patients. Five patients were confirmed to be neo-RAS WT at the time of progression after the first-line treatment. Conversely, one patient who was classified as neo-RAS-WT at the time of first-line progression reacquired NRAS A146T mutation, originally detected in primary tumor tissue. Consistently, before starting the second-line treatment, 30 patients were found to be RAS-mutated in plasma.

Among these 30 patients, 12 (40%) switched to RAS-WT, including 7 (23%) classified as neo-RAS-WT, and 5 (8%) non-shedding. In the seven patients who converted to neo-RAS-WT in the course of the second-line treatment, the initial RAS-mutation became undetectable at 3 months in six patients (86%) and at 6 months in one patient (14%). Combining the patients classified as neo-RAS-WT before starting the second line and those who converted to neo-RAS-WT during the second line, the percentage of neo-RAS-WT cases was 12 out of 35 (34%).

Eleven patients had sustained neo-RAS-WT status that was confirmed at progression, while in one case, the neo-RAS-WT window was transient and the patient reacquired RAS mutation at the time of treatment failure. The median duration of neo-RAS-WT window in the course of the second-line treatment was 5.6 months (range 3–7 months).

The loss of RAS mutation in non-shedding was always transient since all patients reacquired RAS mutation at the time of disease progression, with return of the same variant in all cases (100%). The median duration of RAS-WT window in non-shedding patients was 2.8 months (range: 2–3). Finally, the proportion of persistent RAS mutant patients progressively increased in course of second-line compared to first (18 patients, 52%). At the time of progression from second-line treatment, 31% of patients were neo-RAS-WT and 69% RAS-mutant (Table 1).

### 3.4. Longitudinal Monitoring of RAS Status in Plasma ctDNA During Third-Line Treatment

Before starting the third-line treatment, a blood draw was performed in all patients. At the baseline of third-line treatment, 2 patients were found neo-RAS-WT (as at the time of progression to second-line), 3 were non-shedding and 30 were RAS-mutant.

Of these last, three (10%) switched to RAS-WT during third-line treatment, with only one patient being neo-RAS-WT. Combining the patients classified as neo-RAS-WT before starting the third-line treatment with those who converted to neo-RAS-WT during the course of the third-line, the overall percentage of neo-RAS-WT cases during the third-line treatment was 8.5%. The two patients who were neo-RAS-WT at baseline persisted as neo-RAS-WT for a transient period, reacquiring RAS-mutation at the time of disease progression. The median duration of neo-RAS-WT window in the course of the third-line treatment was 4 months (range 3–6 months). The loss of RAS mutations in non-shedding was always transient since all patients reacquired RAS mutation in plasma at the time of disease progression, with return of the same variant in all cases. The median duration of RAS-WT window in non-shedding was 4 months (range: 3–6 months).

The proportion of persistent RAS-mutant patients progressively increased (29/35, 83%). At the time of progression from the third-line treatment, 6% of patients were neo-RAS-WT, 8% non-shedding and 86% RAS-mutant (Table 1).

The median duration of RAS-WT window in neo-RAS-WT compared to non-shedding is illustrated in Table 2.

Finally, the neo-RAS population had a longer RAS-WT window compared to non-shedding (odds ratio [OR]: 1.23; 95% CI: 1.01–1.50; *p* = 0.037) (Figure 1).

### 3.5. Survival Analysis

Patients who converted to RAS-WT status in ctDNA (both neo-RAS WT and non-shedding) experienced longer progression-free survival (PFS) compared to those with persistent RAS-mutant status during first-line treatment. Specifically, the median PFS (mPFS) for neo-RAS WT patients was 12.0 months (95% CI: 9.12–14.88), while non-shedding patients had an mPFS of 13.4 months (95% CI: 10.92–15.07). Conversely, the persistent mutant group had an mPFS of 11.08 months (95% CI: not reached [NR]-NR) (*p* = 0.004) (Figure 2).

A similar trend was observed in patients receiving second-line treatment, where both neo-RAS-WT patients and non-shedding had an mPFS of 8 months (95% CI: 6.93–9.06 for neo-RAS; 95% CI: not reached for non-shedding). Persistent RAS-mutant patients had an mPFS of 5 months (95% CI: 4.82–5.17) (*p* < 0.0001) (Figure 3).

In the third-line treatment, both neo-RAS-WT patients and non-shedding had an mPFS of 5 months (95% CI: NR for both), while persistent-RAS-mutant patients had an mPFS of 3 months (95% CI: 2.47–3.53); (*p* = 0.001) (Figure 4).

As expected, patients with a longer ctDNA-WT window experienced a better PFS during first-line (hazard ratio [HR]: 0.78; 95% CI: 0.69–0.89; *p* < 0.0001) and second-line treatments (HR: 0.66; 95% CI: 0.52–0.84; *p* = 0.001) (Figure 5).

Furthermore, the duration of RAS-WT window was associated with longer overall survival (OS) in univariate analysis (HR: 0.90; 95% CI: 0.81–0.99; *p* = 0.04). This positive correlation was confirmed through multivariate analysis, adjusted by ctDNA status (neo-RAS WT vs. non-shedding), number of metastatic sites of disease and treatment line at the time of RAS conversion (HR: 0.83; 95% CI: 0.72–0.95; *p* = 0.008) (Figure 6).

Finally, the presence of RAS-mutation in exon 2 in tissue sample at baseline was correlated with a lower probability of conversion to neo-RAS WT ctDNA status (OR: 0.16; 95% CI: 0.03–0.82; *p* = 0.022) (Figure 7).

## 4. Discussion

The introduction of ctDNA analysis has recently allowed for a comprehensive overview of spatial and temporal heterogeneity in mCRC [15].

In RAS-mutant mCRC, the disappearance of RAS mutations in ctDNA over the course of the disease is currently known as “neo-RAS-WT”. This phenomenon is reported to occur in a highly variable range of cases according to previous studies [6]. Importantly, the sudden disappearance of a driver oncogene mutation in ctDNA can suggest two conflicting clinical situations. The first is a therapeutic response since the loss of the mutation in ctDNA might indicate shrinkage of the tumor mass due to effective treatment, followed by a decrease in ctDNA release into the bloodstream. The second scenario is a consequence of tumor evolution under therapeutic pressure; oncological treatments may lead to the emergence of subclones with different genetic profiles [16,17]. Consistent with this last hypothesis, studies comparing tissue biopsies from primary and metastatic sites have demonstrated a reduction in the variant allele frequency of RAS mutations or a full loss of RAS mutations in bevacizumab-pretreated patients [18]. These data suggest that anticancer therapy might exert pressure on RAS-mutant cells, causing a detectable change in the status of ctDNA. Since the first situation is characterized by a lack of ctDNA shedding, whereas the second situation is not, proper confirmation of the presence of ctDNA in plasma samples is needed to address the debated issue of whether neo-RAS-WT might be actionable with EGFR blockade. Consistently, while the term ’neo-RAS-WT’ should strictly refer to cases of RAS-WT detected in plasma with simultaneous evidence of ctDNA as a positive control for tumor content detection [14,17,18,19], many studies are significantly biased by the lack of proper ctDNA confirmation tests. This oversight leads to confusion about the precise meaning of the term ’neo-RAS-WT’.

The clinical implications of this confusion are relevant because only neo-RAS-WT might be resensitized to EGFR blockade, as described in several case reports, unexpectedly demonstrating long-term disease control with salvage regimens containing EGFRi mAbs in patients with neo-RAS-WT in ctDNA [6].

In the GOZILA study, 478 patients with tissue *RAS* MT mCRC were eligible. Neo-*RAS*-WT was defined as the absence of detectable *RAS* MT in plasma and it was assessed in all eligible patients and in a subgroup with at least one somatic alteration detected in plasma. Overall, 1/6 and 2/6 patients with neo-*RAS*-WT treated with EGFRi showed partial response and stable disease for ≥6 months, respectively [7].

In this context the C-PROWESS and Convertix trials, two prospective multicenter, single-arm, phase II trials evaluating the efficacy and safety of panitumumab combined with irinotecan-based chemotherapy in neo-RAS-WT mCRC patients, will probably answer this clinical question [8,9].

In addition to technical issues, various confounding factors may account for the significant differences in the reported incidence of the neo-RAS-WT phenomenon. One critical aspect concerns the high heterogeneity of treatment lines in different study populations [12,20,21]. Uniquely, the present study enabled the full tracking of neo-RAS-WT and the duration of the new-RAS-WT window over three different treatment lines in a prospective manner. We also tracked persistent mutant phenotypes and non-shedding patients, offering hypothesis-generating evidence of how tumor genomics may be altered over the course of different treatments in RAS-mutant mCRC. In accordance with what has been published over the years by our research group [13,22], the percentage of neo-RAS-WT is highest in the second-line treatment, whereas the percentage of non-shedding and persistent mutant has the opposite trend, with the first decreasing and the second increasing in rate over the course of treatment lines, suggesting a rebound in detection as treatments become less effective. Unlike those reported in the literature, 8.5% of patients had mutation loss during third-line therapy, indicating that clonal evolution might persist under different treatment pressures. Some evidence has shown that neo-RAS-WT is a transient phenomenon, with the same RAS-mutant variant re-emerging by the time of disease progression. This lack of durability of the “RAS-WT window” raised the question of whether the neo-RAS-WT can truly represent a clinically useful therapeutic target.

In our study, most neo-RAS-WT patients had a sustained neo-RAS-WT status that could be confirmed at the time of disease progression independent of treatment line, with the exception of a few patients who reacquired RAS mutation at the time of treatment failure. Conversely, the loss of RAS mutations in non-shedding was always transient since all patients reacquired RAS mutation at the time of disease progression, with a return of the same variant in all patients. These results further highlight the importance of correctly discriminating non-shedding from neo-RAS-WT patients. Regardless of the durability of the WT window, we believe that it identifies a population of patients in whom the introduction of second-line EGFRi might represent a therapeutic option. Extending the continuum of care through the delivery of as many lines of efficacious therapy as possible is a desirable goal in RAS-mutant mCRC so that patients with progressive disease might have multiple potentially beneficial treatment options to explore before transitioning into palliative care. Consistently, some encouraging results regarding the use of EGFRi in neo-RAS-WT patients have been published by our group and others, demonstrating a sustained PFS (5–14 months) of patients receiving EGFRi monotherapy alone or in combination with irinotecan for neo-RAS-WT [21,22,23,24,25,26,27,28].

## 5. Conclusions

In conclusion, despite this exploratory study needing to be confirmed by a larger series, it represents the first prospective tracking of neo-RAS-WT phenotypes and the duration of the neo-RAS-WT window across three different lines of treatment. We demonstrate that while the RAS-WT window in non-shedding patients is indeed transient, that observed in neo-RAS-WT is durable and persists until disease progression, highlighting the importance of correctly interpreting neo-RAS-WT in ctDNA and revealing a new perspective for EGFRi efficacy in these patients.

## Figures and Tables

**Figure 1 cancers-17-01070-f001:**
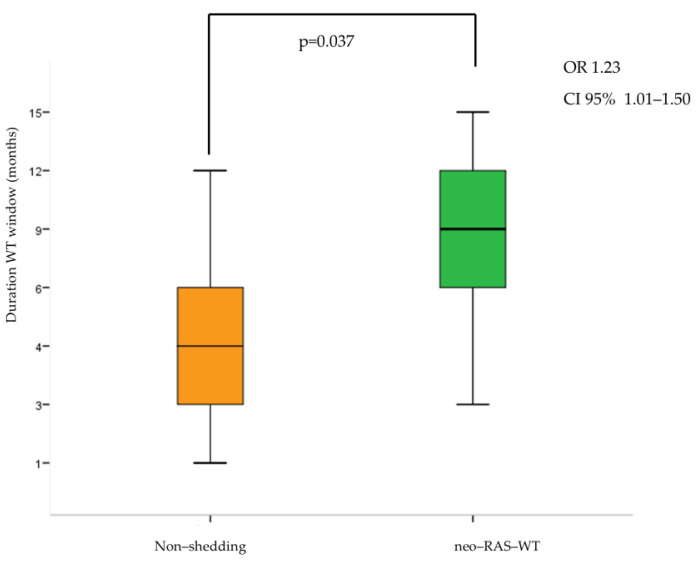
Relation between ctDNA wild-type (WT) status (neo-RAS vs. non-shedding) and duration of WT window in patients with ctDNA WT conversion. OR = odds ratio.

**Figure 2 cancers-17-01070-f002:**
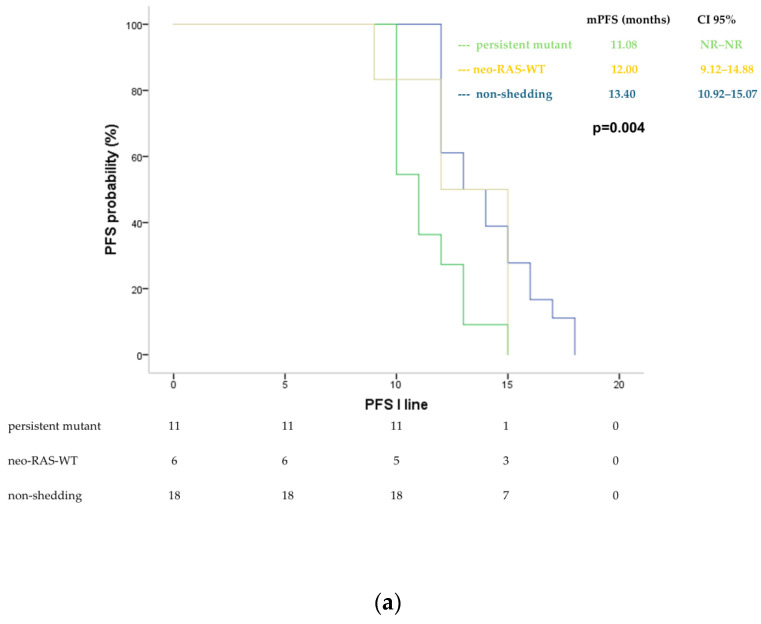
Progression-free survival (PFS) in neo-RAS-WT, non-shedding and persistent mutant patients in the course of first-line treatment illustrated in KM curves (**a**) and swimmer plot (**b**). PD = progression disease.

**Figure 3 cancers-17-01070-f003:**
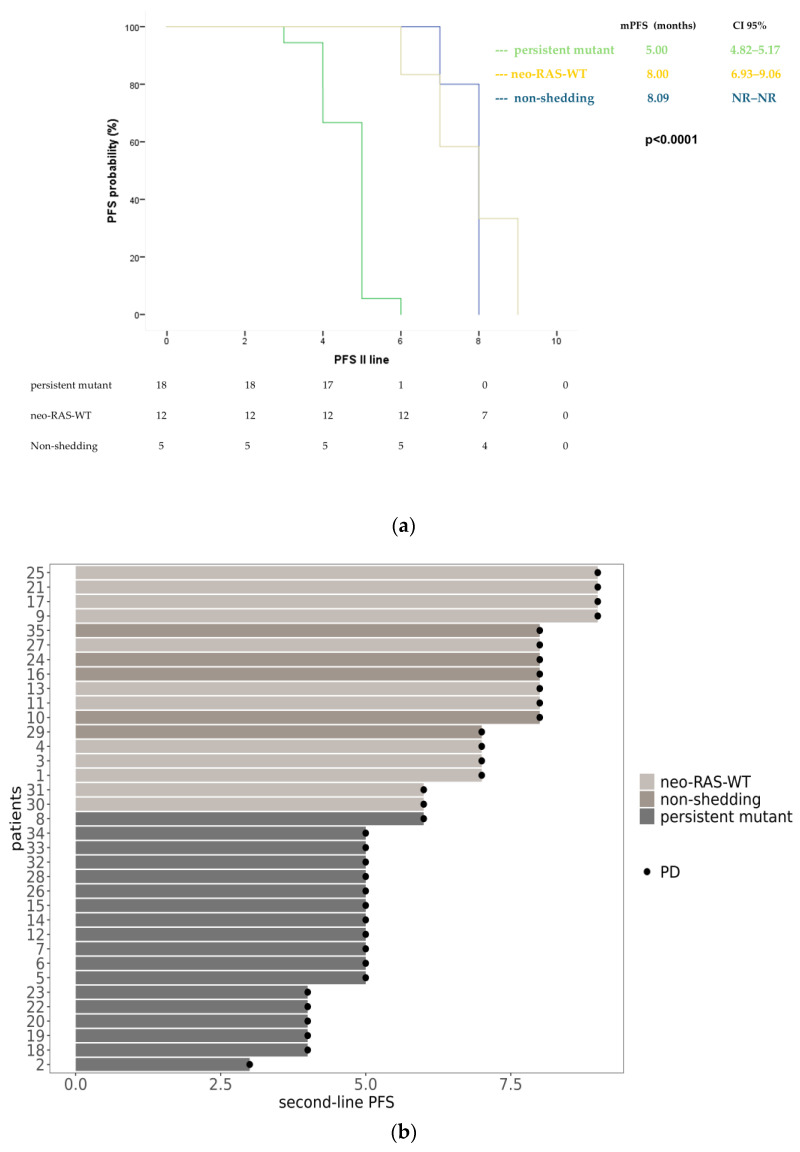
Progression-free survival in neo-RAS-WT, non-shedding and persistent mutant in the course of second-line treatment displayed through Kaplan–Meyer curves (**a**) and swimmer plot (**b**). NR = not reached PD = Progression Disease.

**Figure 4 cancers-17-01070-f004:**
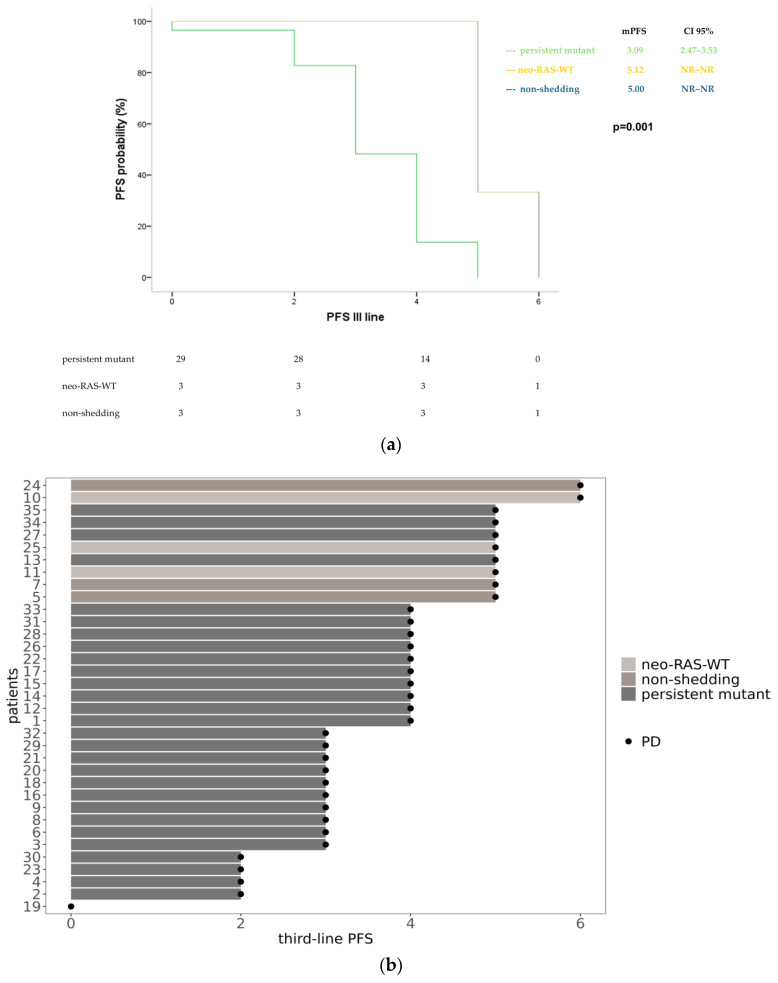
Progression-free survival in neo-RAS-WT, non-shedding and persistent mutant in course of third-line treatment displayed in Kaplan–Meyer curves (**a**) and swimmer plot (**b**) NR = not reached; PD = progression disease.

**Figure 5 cancers-17-01070-f005:**
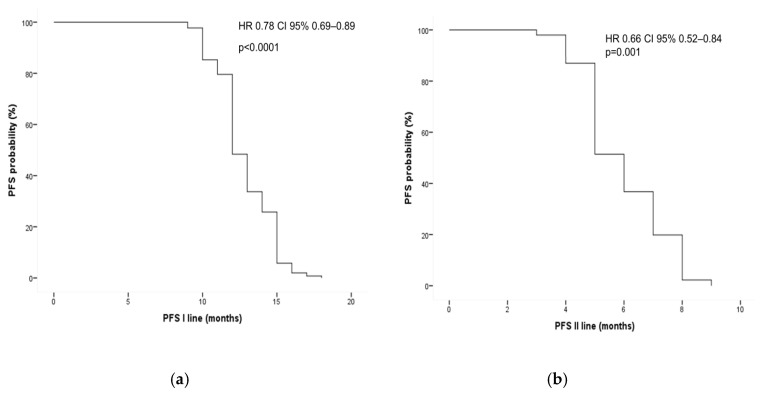
Univariate analysis of progression-free survival in first- (**a**) and second-line (**b**) treatments according to duration of wild-type (WT) window. HR = hazard ratio.

**Figure 6 cancers-17-01070-f006:**
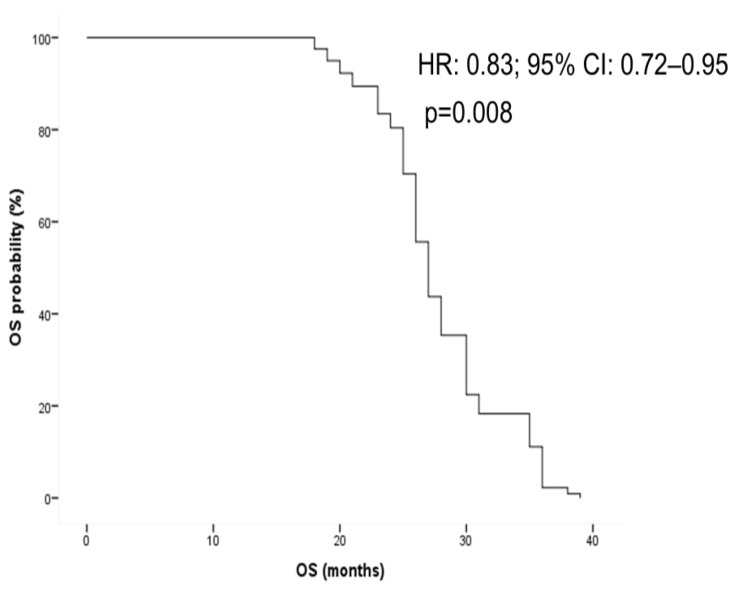
Overall survival according to wild-type (WT) window duration adjusted through multivariate analysis by ctDNA status (neo-RAS WT vs. non-shedding), number of metastasis and line of treatment. HR = hazard ratio; OS = overall survival.

**Figure 7 cancers-17-01070-f007:**
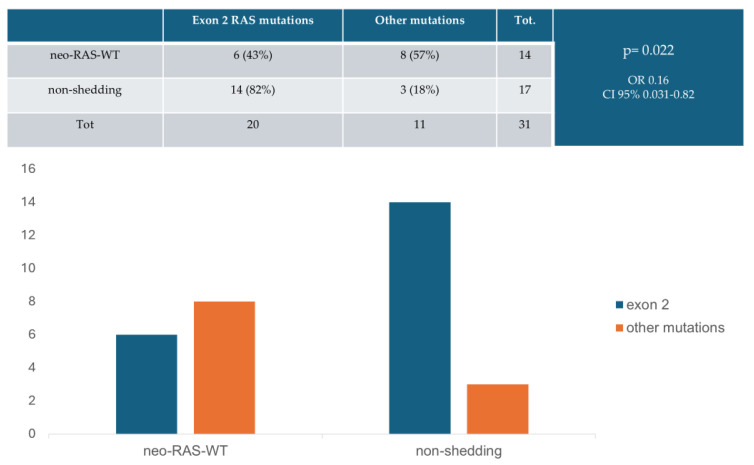
Relationship between ctDNA wild-type (WT) status (neo-RAS WT vs. non-shedding) and tissue RAS-mutation at baseline. OR = odds ratio.

**Table 1 cancers-17-01070-t001:** Percentage of neo-Ras WT, non-shedding and persistent mutant during first, second and third treatment lines.

	Neo-RAS-WT	Non-Shedding	Persistent Mutant
First-line	6 (17%)	18 (51%)	11 (32%)
PD (first-line)	6 (17%)	0 (0%)	29 (83%)
Second-line	12 (34%)	5 (14%)	18 (52%)
PD (second-line)	11 (31%)	0 (0%)	24 (69%)
Third-line	3 (8.5%)	3 (8.5%)	29 (83%)
PD (third-line)	2 (6%)	3 (8%)	30 (86%)

**Table 2 cancers-17-01070-t002:** Median duration of neo-RAS-WT window in neo-RAS-WT vs. non-shedding. PD = progression disease.

	Neo-RAS-WT	Non-Shedding
First line	5.5 months (3–9)	4.7 months (1–9)
Second line	5.6 months (3–7)	2.8 months (2–3)
Third line	4 months (3–6)	4 months (3–6)

## Data Availability

Data sharing is not applicable to this article.

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
