# Peer review of "Evolution of Neo-RAS-WT in Circulating Tumor DNA from First-Line to Subsequent Therapies in Metastatic Colorectal Cancer"

_cancers, 2025, doi:10.3390/cancers17071070_

Round 1
Reviewer 1 Report
Comments and Suggestions for Authors
In the manuscript titled "Evolution of Neo-RAS-WT in Circulating Tumor DNA from First-Line to Subsequent Therapies in Metastatic Colorectal Cancer," the authors evaluate the role of longitudinal profiling by liquid biopsy in monitoring changes in RAS mutational status among patients with tissue-RAS mutant colorectal cancer who are also RAS-mutant at baseline ctDNA assessment. They identify three distinct groups: Neo-RAS-WT, non-shedding, and persistent RAS-mutant.
The topic is of significant interest in the field. However, several concerns need to be addressed:
- Plain summary: The language should be simplified for better accessibility.
- Table 1: Absolute numbers should be included.
- The definitions of Neo-RAS-WT, non-shedding, and persistent RAS-mutant should be uniformly applied across the text, tables, and figures.
- mPFS and other numerical data should be carefully checked for accuracy (Decimal values)
- Typographical errors: For example, in Figure 4, "persistetn" should be corrected.
- Given the limited sample size, the use of Kaplan-Meier curves should be reconsidered. A more informative visualization should be considered, such as a swimmer plot incorporating PFS and conversion time data.
- univariate/multivariate analysis should include significant clinical characteristics
- Clinical characteristics: The manuscript lacks critical information on patient demographics and clinicopathological features, including primary tumour status, tumour burden, number of metastatic sites, and site of progression.
- The manuscript should specify the type of RAS mutations and their corresponding MAF values.
- A brief review of existing data on the use of anti-EGFR therapy in the Neo-RAS-WT setting should be added.
- All abbreviations should be consistently spelt and formatted throughout the manuscript.
No further comment
Reviewer 2 Report
Comments and Suggestions for Authors
The work "Evolution of Neo-RAS-WT in Circulating Tumor DNA from 2 First-Line to Subsequent Therapies in Metastatic Colorectal 3 Cancer", presented by Professor Gazzaniga and co-workers is very interesting and innovative. Reversion of RAS mutations in cancer opens new therapeutical options and, as this can lead to a better overall survival of cancer patients, it must be studied. However, there are some flaws in Professor Gazzaniga's work, being the most limiting the number of patients included in the study. Some of the groups are too small to give a real picture of the status of those patients, and this should be corrected before the publication of this kind of data.
Other points that should be addressed before publication are:
1) The text referring to Figure 1 and the data shown in Figure 1 are contradictory. In the text the authors comment: "Finally, the neo-RAS population had a longer RAS-WT window compared to non shedding (Odds Ratio [OR]: 1.23; 95% CI: 1.01–1.50; p=0.037) (Fig. 1)", but when you look at Figure 1, the non-shedders show a duration of WT window longer than neo RAS WT.
2) Figures 5 and 6. The authors show several Kaplan-Meyer graphs analyzing PFS and OS and give a statistical value for those analyses, but in the graphs only one condition is depicted. Shouldn't these graphs include the group (control/non-reversed mutant RAS) used to obtain the statistical value in the comparison?
Round 2
Reviewer 2 Report
Comments and Suggestions for Authors
The authors have made the appropriate changes to the manuscript and now is ready for publication.